# What Contributes to Athlete Performance Health? A Concept Mapping Approach

**DOI:** 10.3390/ijerph20010300

**Published:** 2022-12-24

**Authors:** Erin A. Smyth, Alex Donaldson, Michael K. Drew, Miranda Menaspa, Jennifer Cooke, Sara A. Guevara, Craig Purdam, Craig Appaneal, Rebecca Wiasak, Liam Toohey

**Affiliations:** 1Research Institute for Sport and Exercise, Faculty of Health, University of Canberra, Bruce Campus, 11 Kirinari Street, Bruce, Canberra, ACT 2617, Australia; 2Centre for Sport and Social Impact, La Trobe University, Melbourne, VIC 3086, Australia; 3Australian Institute of Sport, Bruce, Canberra, ACT 2617, Australia; 4School of Physiotherapy, University of Canberra, Bruce, Canberra, ACT 2617, Australia; 5La Trobe Sport and Exercise Medicine Research Centre, Melbourne, VIC 3086, Australia; 6Research School of Psychology, Australian National University, Canberra, ACT 2601, Australia

**Keywords:** athletic performance, sports, athletes, health, injury

## Abstract

Background: In high-performance sport, athlete performance health encompasses a state of optimal physical, mental, and social wellbeing related to an athlete’s sporting success. The aim of this study was to identify the priority areas for achieving athlete performance health in Australia’s high-performance sport system (HPSS). Methods: Participants across five socioecological levels of Australia’s HPSS were invited to contribute to this study. Concept mapping, a mixed-methods approach incorporating qualitative and quantitative data collection, was used. Participants brainstormed ideas for what athlete performance health requires, sorted the ideas into groups based on similar meaning and rated the importance, and ease of achieving each idea on a scale from 1 (not important/easiest to overcome) to 5 (extremely important/hardest to overcome). Results: Forty-nine participants generated 97 unique statements that were grouped into 12 clusters following multidimensional scaling and hierarchical cluster analysis. The three clusters with highest mean importance rating were (mean importance rating (1–5), mean ease of overcoming (1–5)): ‘Behavioral competency’ (4.37, 2.30); ‘Collaboration and teamwork’ (4.19, 2.65); ‘Valuing athlete wellbeing’ (4.17, 2.77). The 12 clusters were grouped into five overarching domains: Domain one—Performance health culture; Domain two—Integrated strategy; Domain three—Operational effectiveness; Domain four—Skilled people; Domain five—Leadership. Conclusion: A diverse sample of key stakeholders from Australia’s HPSS identified five overarching domains that contribute to athlete performance health. The themes that need to be addressed in a strategy to achieve athlete performance health in Australia’s HPSS are ‘Leadership’, ‘Skilled people’, ‘Performance health culture’, ‘Operational effectiveness’, and ‘Integrated strategy’.

## 1. Introduction

Many factors contribute to successful athletic performance including technical, tactical and physical attributes alongside the athlete–coach relationship [1]. Athletic performance is intrinsically linked with athlete health. Injuries and illnesses have been consistently demonstrated to impair team and individual success [2], where pre-competition [3] and in-competition [4] injuries are negatively associated with achieving key performance indicators. Conversely, an athlete’s ability to consistently train and compete without interruption is a key factor for achieving performance success as evidenced across many sports including track and field [5], football [6], and basketball [7].

Within the context of Australia’s high-performance sport system (HPSS), the World Health Organization’s definition of health [8] has been adapted to define athlete performance health as *a state of optimal physical, mental, and social wellbeing related to an athlete’s sporting success, and not merely the absence of illness or injury that limits participation*. Athlete health was recently identified as the most important and feasible issue to address for athlete retention in the Australian high-performance pathway system [9]. In addition to the impact health has on athlete performance and attrition, most importantly, athletes are people first, where wellbeing and safety should be placed at the forefront of their athletic journey [10,11,12,13]. Therefore, athlete health must be prioritized in high-performance sport.

The complex multidimensional nature of athlete health is influenced by many proximal and distal factors within a socioecological framework [14]. In sport, these factors are often considered in isolation through a reductionist perspective; however, to optimize and protect athlete health we must move beyond simplistic monocausal thinking to consider the complexity and multifactorial nature of the whole system [15]. Hanson et. al. [16] illustrated this well with ‘the injury iceberg’ concept, demonstrating that athlete intrapersonal physical factors such as lumbar range of movement are merely the ‘tip’ of the iceberg and other interpersonal, organizational, community, and societal factors can have a greater influence over the prevention and occurrence of injury and illness. 

A complex system framework’s fundamental premises that the parts of a system are inter-related, and that the objective of the whole system defines the function of each part [17] are required to determine the factors that influence athlete performance health. This study aimed to consult all socioecological levels [18] of Australia’s high-performance sport system to identify the priority domains and areas for gaining a better understanding in how to further develop athlete performance health.

## 2. Methods

### 2.1. Participants

This project used a socioecological framework [18] to systematically identify the factors that contribute to achieving athlete performance health. The five socioecological levels identified for the purposes of this study were (i) current Australian HPSS athletes; (ii) Australian HPSS support staff (e.g., coaches, physiotherapists, dietitians, and strength and conditioning coaches); (iii) National Sporting Organization (NSO) management staff; (iv) middle management staff of the National Institute Network (NIN)—a collective of the nine Australian state institutes of sport and state academies of sport; (v) Australian Institute of Sport (AIS) and NIN Directors. The socioecological level of each participant was determined by the research team.

Purposive sampling was used to identify potential study participants from each of the five socioecological levels in the Australian HPSS. To be eligible, athlete participants were required to be either a pre-elite or elite athlete, and non-athlete participants’ work needed to influence the performance health management of a pre-elite or elite athlete. Potential participants were sent an email by AIS personnel who were routinely in contact with each population. Athletes were offered a $200 gift voucher incentive for participating. It was assumed other participants would contribute during employed hours so no compensation for their time was offered. Ethics approval was obtained from the Australian Institute of Sport Human Research Ethics Committee (approval number: 20211001).

### 2.2. Study Design

Concept mapping (CM) is a valid and reliable process [19] that uses mixed methods to generate and organize ideas from a group of individuals around a topic of interest [20]. These ideas are collected from and organized by study participants before being represented visually on a map [21]. Qualitative methods are used to generate the data which are then analyzed using quantitative methods [22]. The CM process involves three key processes: (i) brainstorming and generating “facilitator statements”, (ii) statement sorting and rating unique facilitator statements, (iii) discussing the results with the key stakeholder group [20,21,23]. 

### 2.3. Procedures

Participants completed brainstorming, sorting, and rating activities using the groupwisdom^TM^ online platform (Concept Systems Inc., Ithaca, NY, USA). Following recruitment, participants were asked to complete background questions (Appendix A) such as role, gender identity, sports they were involved in, etc. to aid interpretation of the results. They were then asked to complete the brainstorming activity. Two reminder emails were sent before this step was closed. All participants, even if they did not participate in the brainstorming, were emailed an invitation to participate in the next two tasks (statement sorting and rating). Two reminder emails were also sent before the sorting and rating tasks were closed. Data was collected over six weeks during November and December 2021. The groupwisdom^TM^ platform was used to analyze the data and to generate concept maps and additional displays of the data.

#### 2.3.1. Brainstorming

Participants were asked to write a brief statement (one idea) to complete the following focus prompt: ‘Performance Health is a state of optimal physical, mental, and social well-being related to an athlete’s sporting success, and not merely the absence of illness or injury that limits participation. With this in mind, please complete the statement: Athlete performance health requires……….’. Participants could add as many statements as they desired. The following instruction was used: ‘Please keep each statement brief, just one thought. You may add as many statements as you wish’. Participants were encouraged to add multiple statements and review the statements made anonymously by other participants to check if an idea had already been contributed and to stimulate new ideas.

#### 2.3.2. Statement Synthesis

After the brainstorming phase was closed, eight reviewers (ES, MD, LT, RW, JC, CP, AD, SG) synthesized and edited the brainstormed statements to ensure each statement was relevant to the project focus, contained a unique idea (i.e., each idea was represented once), and was clearly expressed. This iterative process involved: (i) identifying and deleting statements not related to the focus prompt; (ii) identifying and splitting statements with two or more unique ideas; (iii) identifying and editing statements as required to ensure all members of the group agreed on the essential meaning of the statement; and (iv) identifying statements that represented the same idea, selecting the statement that best represented the idea, and deleting the other similar statements. Synthesizing and editing occurred in one meeting and follow up emails until consensus was reached among the reviewers. The participants’ original voice was preserved where possible [21]. The resulting list of synthesized statements was made available to participants to complete the subsequent sorting and rating process.

#### 2.3.3. Statement Sorting

Using the groupwisdom^TM^ online platform, participants sorted the synthesized statements into ‘piles’. They were instructed to sort the statements according to similarity in meaning and to name each pile based on its theme. Participants were asked not to create piles of unrelated ideas (e.g., ‘miscellaneous’) or create piles according to priority (e.g., ‘hard to do’). They were informed that the number of piles people create varies, and 5 to 15 piles was recommended as a guideline to organize the statements [19].

#### 2.3.4. Statement Rating

Participants were asked to rate each of the synthesized statements on a five-point Likert scale by answering the following questions: ‘*On a scale from 1 (not at all important) to 5 (extremely important), how important is this statement to athlete performance health?*’; and ‘*On a scale from 1 (easiest to achieve) to 5 (hardest to achieve), how easy is it for this statement to be achieved?*’.

### 2.4. Analysis

Using the groupwisdom^TM^ online platform, a similarity matrix was developed based on the frequency with which each statement was sorted with every other statement by all participants. Multidimensional scaling was applied to the matrix to position each sorted statement as a separate point on a two-dimensional ‘point map’. The location of each point on the map is an indicator of its relationship to all other points. Points that are further away from each other were less frequently sorted together and points that are closer together were more often sorted together by participants. A stress index was calculated to indicate how well the point map’s configuration represents the original sorting data. A low stress index suggests a better overall fit [24].

Hierarchical cluster analysis was used to develop ‘cluster maps’ by separating the statements on the point map into clusters of related statements based on participants’ aggregated sorting data [24]. To select the number of clusters that best represented the sorted data, cluster maps were produced for a 13-cluster solution through to a 10-cluster solution to find the cluster level that retained the most useful detail between clusters while merging the clusters that seemed to belong together. Once the most appropriate cluster solution was agreed upon by members of the research team (AD, JC, SG, MD, ES), every statement in each cluster was reviewed and, if a statement was considered to be a better conceptual fit in an adjacent cluster (i.e., the theme of an adjacent cluster more accurately and appropriately encompassed the idea underpinning the statement), cluster boundaries were re-drawn [25]. All boundary-redrawing decisions were guided by cross referencing back to the similarity matrix to check how frequently participants had grouped the statement being considered for reassignment with statements in adjacent clusters, and based on consensus among the research team. Once the final cluster map was decided, the research team reviewed the map to determine if it could be further partitioned into overarching domains of related clusters [26].

Descriptive statistics were calculated for ‘importance’ and ‘ease of achieving’ ratings to generate a ‘go-zone graph’ [24] in which the mean ‘importance’ rating for each statement was plotted along the x-axis and the mean ‘ease of achieving’ rating for each statement was plotted along the y-axis. The resulting scatterplot was divided into four quadrants using the grand mean for each scale with the right-hand quadrants representing the two priority areas for action—the bottom quadrant contains relatively important and easy-to-achieve statements that represent short-term easy wins while the top quadrant contains relatively important but harder-to-achieve statements that represent longer-term goals requiring more strategic responses. Action should start in both areas so that stakeholders feel like progress is being made addressing issues in the bottom right quadrant while strategies are being put in place to address the important but hard-to-do things located in the top right quadrant. A Pearson product moment correlation coefficient was calculated to demonstrate the relationship between the importance and ease of achieving a statement [27].

## 3. Results

### 3.1. Participants

Forty-nine participants contributed to CM data—37 brainstormed ideas, 25 sorted statements, 29 rated statements for importance, and 28 rated statements for ease of achieving. Table 1 displays the participants’ characteristics for each CM task. There was relatively equal representation of male and female participants, a high proportion of participants (39%) worked with multiple sports, there were fewer participants representing socioecological Level 4 (8%) and Level 5 (4%), and there was no Aboriginal or Torres Strait Islander representation and limited LGBTQI representation.

### 3.2. Brainstorming

Thirty-seven participants generated 101 statements which, after synthesis and editing, yielded 97 unique and relevant statements for sorting and rating (Table 2).

### 3.3. Sorting: Concept Map

Figure 1 is the cluster map developed following multidimensional scaling and hierarchical cluster analysis. An example of statements frequently grouped together and therefore close to each other on the cluster map are statement 22 (‘*the best practitioners with strong links to coaches*’) and statement 23 (‘*the best practitioners with experience and deep understanding of the sport and athletes*’). These statements were grouped together 20 times. Conversely, statement 37 (‘*expertise to understand data to inform decisions*’) was never grouped with statement 73 (‘*an individualized, flexible and adaptable environment to capitalize on athletes’ strengths and potential (e.g., athlete-specific strength and conditioning programs*)’), so these statements are very far apart on the cluster map. The stress index was 0.33 which indicates a less than 1% probability of the map having no structure [19].

A 12-cluster map was selected as the best representation of the sorted data. Eighteen (19%) statements were reassigned by the authors to adjacent clusters for a better conceptual fit. The statements within each cluster are presented in Table 2, with the domains, clusters, and statements within them organized from most to least important. We identified that the 12 clusters could be further separated into five overarching domains: ‘Leadership’, ‘Skilled people’, ‘Integrated strategy’, ‘Performance health culture’, and ‘Operational effectiveness’. The mean importance and ease of each Domain are also listed in Table 2.

### 3.4. Rating: Perceived Importance and Ease of Achieving

The mean ratings of importance and ease of achieving for each of the 97 statements and for the 12 clusters are provided in Table 2. Cluster 1, ‘*Behavioral competency*’, was deemed to be the most important cluster (4.37) and the cluster that was easiest to achieve (2.30). Cluster 12, ‘*Roles and responsibilities*’, was the lowest-rated cluster in terms of importance (3.56) and ‘*Efficient use of resources*’ (Cluster 5) was found to be the hardest to achieve (3.10).

The most important individual statement was number 70, ‘*clear communication*’ (4.79), and the least important was statement 80, ‘*professionals staying ‘in their lane’ and not trying to be a ‘jack of all trades’’* (2.69). Statement 34, ‘*sufficient financial and human resources*’, was rated the hardest to achieve (3.78) and statement 27, ‘*communication with the athlete involved*’, was deemed the easiest to achieve (1.57).

### 3.5. ‘Go-Zone Graph’: Ideas to Be Prioritized

Figure 2 displays the go-zone graph for all 97 statements. There are 27 statements in the top right quadrant, indicating they were rated above average on both importance and difficulty achieving. It is anticipated that these ideas will require the most investment to achieve. The bottom right quadrant includes statements (n = 27) that were also considered relatively important but less difficult to achieve and could be considered as a starting point to achieve an ‘early win’. The left quadrants include statements that were deemed less important and therefore can be considered a lower priority. The Pearson product moment correlation coefficient for the go-zone graph indicates minimal correlation between ‘Importance’ and ‘Ease of achieving’ (r = −0.19). Therefore, the more important statements were not necessarily deemed to be more difficult or easy to achieve.

### 3.6. Pattern Matching

The mean importance ratings from athletes, performance staff, health staff, and administrative staff for each cluster were compared and displayed in Figure A1 (Appendix B). Four comparisons were found to be significantly different.

## 4. Discussion

The purpose of this concept-mapping project was to identify and prioritize the key elements that stakeholders in the Australian HPSS thought were important for facilitating athlete performance health. We identified five overarching domains to represent the ideas shared by stakeholders in Australia’s high-performance sport system for achieving athlete performance health. The ‘Performance health culture’ domain was identified as the most important and ‘Operational effectiveness’ was identified as the most difficult to overcome. These domains are not unique to athlete performance health in high-performance sport; they have been reported to be key business challenges across a variety of organisations [28]. As demonstrated by the cluster map (Figure 1), leadership is central to athlete performance health and therefore integral to the other domains.

‘Performance health culture’ was thought to be of high importance to achieve athlete performance health. Organizational culture has been described as the sum of peoples’ values and beliefs which guide and shape behaviour [29]. This domain is a good example of organizational culture as it encompasses the clusters of ‘valuing athlete wellbeing’, ‘philosophical beliefs of performance health’, and ‘behavioral competency’. Two-thirds (64%, n = 16) of the ideas included in this domain were rated by participants to have above average importance. Culture has also been shown to be a key component to performance in high-performance sport [30]. This clearly indicates that addressing ‘performance health culture’ is critical for achieving athlete performance health.

Leadership is integral to organizational change [31], and has been defined as ‘a process whereby an individual influences a group of individuals to achieve a common goal’ [32]. Organizational change simply refers to the movement of an organization from one state to another [33] and is essential for the development and implementation of health strategies within a multi-layered sporting system. Half the ideas (n = 3) in the leadership cluster were located in the top right quadrant of the go-zone graph suggesting that they are relatively important but may be difficult to achieve. Given the centrality of the leadership domain in the cluster map and the relative importance placed on leadership by participants and organizational change researchers, leadership should be prioritized when developing and implementing strategies targeting improvement in athlete performance health.

The ‘skilled people’ domain includes three clusters: ‘competency, capacity, and expertise’, ‘roles and responsibilities’, and ‘collaboration and teamwork’. Two thirds (n = 18, 69%) of the ideas included in this domain had above-average ratings for importance. This indicates that Australia’s high-performance sport system places a lot of importance on ensuring coaches and support staff are of the highest quality, have a clear understanding of their own role and the roles of their colleagues, and work together to achieve and maintain athlete performance health. To facilitate this, the high-performance sport industry should consider addressing the issues identified by Dwyer et. al. [34] such as long and unusual work hours, job insecurity, and income disparity to ensure the best people are recruited and retained in the HPSS. Leadership is also key for implementing systems and strategies to support the workforce and encourage collaboration. Addressing these workforce challenges also needs to be prioritized for achieving athlete performance health.

The ‘efficient use of resources’, ‘education and understanding’, and ‘use of data and research’ clusters were incorporated into the ‘operational effectiveness’ domain. This domain had 11 ideas (48%) with above-average ratings for importance. Most (n = 9) of these important ideas were spread across the ‘efficient use of resources’ and ‘education and understanding’ clusters. Both issues are eloquently described by Hanson et. al., [16] who presented the idea of health promotion needing an ecological approach. To achieve a sustainable solution, the system must have access to the resources necessary to maintain the desired outcome and the ability to use the resources (i.e., education). Education around athlete performance health is essential to provide a common language for all, enhancing communication and shared goal setting [35]. 

Interestingly, the clusters within the ‘integrated strategy’ domain (‘integration of performance health and performance outcome’ and ‘health system and strategy’) both had average importance ratings below the grand mean for importance rating. This domain had the least number of ideas (n = 6, 35%) rated above the overall average importance when compared to the other domains. It has been reported that there can be conflict between performance staff (i.e., coaches and strength and conditioning coaches) and health staff (i.e., physiotherapists and doctors) when managing the health of an elite athlete [36]. The relatively low importance ratings for the ideas included in the ‘integration of performance health and performance outcome’ cluster in this study suggests that participants—all stakeholders in Australia’s high-performance sport system—do not currently perceive this conflict, if it exists, to be a priority for attention when considering athlete performance health. This lack of perceived conflict is supported by there being no significant difference in the mean importance rating for this cluster given by 11 health professionals when compared to the mean importance rating given by the five performance staff who completed the rating task (Appendix B).

The three clusters with the highest mean importance rating were ‘behavioral competency’, ‘collaboration and teamwork’, and ‘valuing athlete wellbeing’. Ideas with above-average importance ratings within these three clusters include honesty, a supportive culture, trust, collaboration across the entire support team and coach, athlete engagement, and safety. Supporting and developing these ideas does not require financial support. Rather it requires an organizational culture and environment that fosters and values these behaviors. In another CM project conducted by our research team, stakeholders across all socioecological levels of an Australian state institute of sport were asked why athletes leave the high-performance pathway system before medaling at a pinnacle international senior competition. ‘Poor injury management’, ‘psychological abuse’, and ‘compromised psychological wellbeing’ were amongst the most important reasons for why athletes leave the HPSS [9]. The traits identified in our three most important clusters (‘Behavioral competency’, ‘Collaboration and teamwork’, ‘Valuing athlete wellbeing’) address these identified reasons for leaving the HPSS. The ideas in these clusters need to be prioritized when recruiting workforce for Australia’s HPSS to underpin athlete performance health and to improve athlete attrition rates.

It is important that the strengths and limitations of this study are acknowledged and considered when interpreting the findings. The use of concept mapping is a strength as it is a real-world practical consultative method that involves online qualitative structured group data collection from a geographically dispersed cohort and then applies quantitative tools to develop visual maps representing the relationships between the groups’ ideas [24]. This process enabled us to learn from participant’s individual experiences and gain valuable quantitative data to inform the development of a national framework for athlete performance health. 

Limitations of this study are the lack of representation of minority groups and limited representation of level 4 and 5 socioecological levels. We set out to obtain equal representation of male and female participants and representation from minority groups such as indigenous, people with a disability, and LGBQTI. We achieved close to equal representation for the sexes with 53% (n = 26) of participants identifying as male, 45% (n = 22) as female, and 2% (n = 1) as non-binary. We had no indigenous representation despite the Aboriginal/Torres Strait Islander community making up 3.2% of Australia’s population as of 2021 [37]. Three (6%) participants identified as LGBQTI which is below the 11% of Australia’s population in 2014 as reported by the Australian Human Rights Commission [38]. We had four (8%) participants representing paralympic sport—it is difficult to determine if this is a fair representation because it is unknown how many people in Australia’s HPSS currently work with paralympic sport. Future research is needed to assess the current demographic of Australia’s HPSS to ensure there is adequate diversity across the workforce who are well equipped to nurture Australia’s sporting talent. 

An additional limitation to be aware of when the international community are interpreting these results is that this study is very specific to the Australian sports culture but could also be reproduced in other high-performance sport systems.

## 5. Conclusions

This study identified the following themes necessary to achieve athlete performance health from the perspective of stakeholders in Australia’s HPSS: ‘Leadership’, ‘Skilled people’, ‘Performance health culture’, ‘Operational effectiveness’, and ‘Integrated strategy’. To address these themes, we identified items that sporting organizations can incorporate in their sport strategies and operational plans to maximize the health and performance of their athletes.

## Figures and Tables

**Figure 1 ijerph-20-00300-f001:**
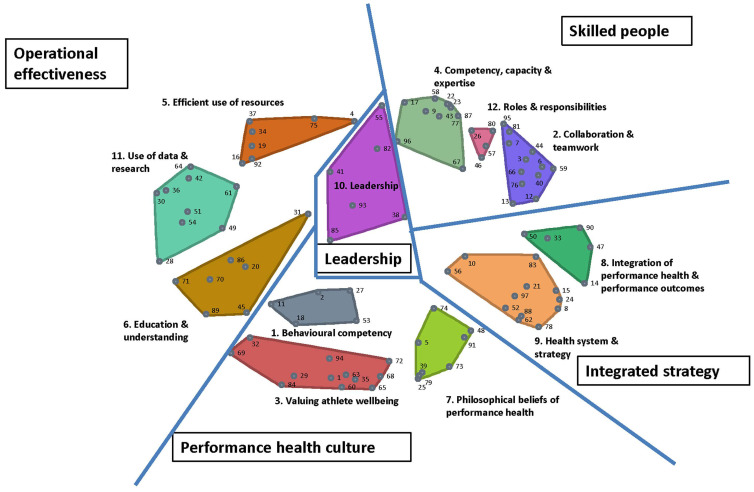
Five-domain, 12-cluster map of ideas from members of the high-performance sport system for achieving athlete performance health.

**Figure 2 ijerph-20-00300-f002:**
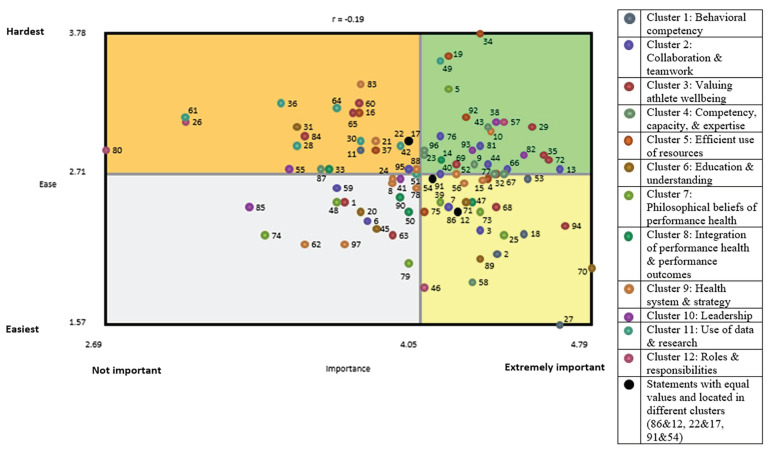
Go-zone graph of ideas from members of Australia’s high-performance sport system for achieving athlete performance health.

**Table 1 ijerph-20-00300-t001:** Participant demographic summary and socioecological level and role for each CM task.

	Overall N (%)	Brainstorming	Sorting	Importance Rating	Ease Rating
**Gender**					
Male	26 (53%)	21	12	12	11
Female	22 (45%)	15	12	16	16
Non-binary	1 (2%)	1	1	1	1
**Type of sport**					
Winter Olympic Team	3 (6%)	2	0	2	2
Winter Olympic Individual	1 (2%)	1	1	1	1
Summer Olympic Team	14 (29%)	9	7	9	9
Summer Paralympic Team	4 (8%)	4	1	2	2
Summer Olympic Individual	4 (8%)	3	3	3	3
Multiple Sports	19 (39%)	14	12	11	10
Commonwealth Games only	3 (6)	3	1	1	1
Professional	1 (2%)	1	0	0	0
**Socioecological level/role**					
**Level 1**					
Athlete	11 (22%)	10	5	7	7
**Level 2**					
Physiotherapist	11 (22%)	10	6	7	7
Coach	3 (6%)	2	1	2	2
S&C coach	1 (2%)	1	1	1	1
Psychologist	2 (4%)	1	2	1	1
Dietitian	1 (2%)	1	0	0	0
AW&E provider	1 (2%)	1	0	0	0
Medical officer	1 (2%)	0	1	0	0
AIS research/technical roles	5 (10%)	0	4	5	5
**Level 3**					
Performance support manager	3 (6%)	3	1	1	1
High performance manager	4 (8%)	3	2	2	2
**Level 4**					
AIS manager	4 (8%)	4	2	2	2
**Level 5**					
Director	2 (4%)	1	0	1	0
**Aboriginal/Torres Strait Islander**					
Yes	0	0	0	0	0
No	49 (100%)	37	25	29	28
**LGBTQI**					
Yes	3 (6%)	2	2	2	2
No	46 (94%)	35	23	27	26
**TOTAL**	**49**	**37**	**25**	**29**	**28**

N, number; AIS, Australian Institute of Sport; AW&E, Athlete wellbeing and engagement; LGBTQI, lesbian, gay, bisexual, transgender, queer (or questioning), and intersex.

**Table 2 ijerph-20-00300-t002:** Statements generated during the concept-mapping brainstorming process including the domain and cluster in which each statement fits, mean importance and ease of overcoming ratings, and go-zone graph quadrants for each statement. Importance: 1 = Not at all important, 5 = Extremely important. Ease: 1 = Easiest to achieve, 5 = Hardest to achieve. Top right quadrant: above mean importance that are the hardest to achieve (green). Bottom right quadrant: above mean importance that are easiest to achieve (gold).

		Mean Rating		Go-Zone Graph Quadrant
		Importance	Ease	
**‘Performance health culture’ Domain**		**4.19**	**2.52**	
**Cluster 1: Behavioral competency**		**4.37**	**2.30**	
27	communication with the athlete involved	4.66	1.57	
53	a supportive culture	4.52	2.68	
18	honesty	4.50	2.26	
2	the basics to be done well	4.38	2.11	
11	self-awareness	3.79	2.89	
**Cluster 3: Valuing athlete wellbeing**		**4.17**	**2.77**	
94	athlete engagement	4.68	2.32	
72	a physically, mentally, emotionally, culturally safe training environment	4.61	2.82	
35	athletes to feel safe to discuss how they are feeling and coping—physically, mentally, and socially	4.59	2.86	
29	athletes to feel safe to talk about mental health without negative consequence	4.54	3.07	
68	planning and management for athletes in periods of poor mental health	4.38	2.46	
32 *	coaches to have an open mind about wellbeing	4.38	2.71	
69 *	athlete’s mental health to be monitored regularly, leading to a mental health plan when required	4.21	2.79	
63	an understanding of what the athlete wants to work towards when determining key performance indicators	3.93	2.25	
60	a balance of life and sport demands	3.79	3.25	
65	a well balanced lifestyle	3.76	3.18	
1	consistent day to day efforts in training the body and mind	3.72	2.50	
84	a sustainable balance that encapsulates self-awareness of multiple identities e.g., Student/Athlete/Young person/Community member	3.55	3.00	
**Cluster 7: Philosophical beliefs of performance health**		**4.03**	**2.50**	
25	an approach that is “athlete centered”	4.41	2.25	
73	an individualized, flexible, and adaptable environment to capitalize on athletes’ strengths and potential (e.g., athlete-specific strength and conditioning programs)	4.31	2.43	
5	shifting the culture within and across sport from ‘win-at-all-costs’, to performance built on a platform of health	4.17	3.36	
39	understanding and responding to the effects of mental or social stress on physical health and performance	4.14	2.50	
91	a balanced view of all the fundamental elements of performance, so that health can be appropriately addressed in the context of performance	4.10	2.68	
79	an understanding of an athlete’s age and life stages	4.00	2.04	
48	a balance of innovation and ‘basics’	3.69	2.50	
74	putting performance at the center	3.38	2.25	
**‘Integrated strategy’ Domain**		**4.00**	**2.66**	
**Cluster 8: Integration of Performance Health and Performance Outcomes**		**4.01**	**2.61**	
47	coaches and support staff to consider physical, mental, and social wellbeing when designing training and rehabilitation programs	4.28	2.50	
14 ^+^	a bespoke, wholistic, collaborative, planned and evidence-based approach in order to tolerate the training stimulus to optimize performance outcomes	4.14	2.82	
50	clear accountabilities and responsibilities as they relate to health and performance	4.00	2.43	
90	health practitioners to strive for performance, not just health	3.96	2.54	
33	recognition that athletes can’t win without a proper high performance health infrastructure	3.66	2.75	
**Cluster 9: Health system and strategy**		**4.00**	**2.70**	
10 ^!^	an effective and functioning health system	4.36	3.04	
15	a system that supports training planning, underpinned by medical and performance support, to optimize athletic adaptations for specific competition outcomes	4.32	2.67	
56 ^!^	prevention programs to be designed and implemented that address the major injuries and illnesses for the sport	4.24	2.64	
52	a clear performance planning process that articulates ‘what it takes to win’ and how the athlete(s) is able to achieve this without being injured or ill.	4.21	2.71	
78	an agreed strategy	4.03	2.61	
88	full integration into the training plan	4.03	2.75	
8	alignment to a performance health strategy	3.93	2.64	
24	a unique approach, as opposed to general population health, that is fit for purpose	3.93	2.68	
21	consideration of the needs of the sport which will result in a unique support structure	3.86	2.96	
83	a whole-of-sport approach	3.79	3.39	
97	a health system with a stated aim	3.72	2.18	
62	having a clear timetable for health check-ins and re-evaluations with ability to adapt as required	3.55	2.18	
**‘Operational effectiveness’ Domain**		**3.99**	**2.85**	
**Cluster 5: Efficient use of resources**		**4.12**	**3.10**	
4 ^>^	access to high quality services	4.34	2.68	
34	sufficient financial and human resources	4.31	3.78	
92	dedicated time to monitor, understand and plan to address major health concerns	4.25	3.14	
19	appropriate resource allocation from leadership (board or management)	4.17	3.61	
75 ^>^	appropriate information at the appropriate time between athlete, coach, and support team	4.07	2.43	
37	expertise to understand data to inform decisions	3.86	2.89	
16	leadership/stewardship in engagement, implementation, evaluation, and refinement	3.79	3.18	
**Cluster 6: Education and understanding**		**4.10**	**2.40**	
70	clear communication	4.79	2.00	
89 ^#^	a clear goal	4.31	2.07	
71	understanding the physical health of female athletes across all ages and life stages; pre-puberty, puberty, conception, pregnancy, postnatal and beyond	4.25	2.50	
86	knowledge of how to plan	4.21	2.43	
45 ^#^	clarity around what performance health is, what it isn’t, and how it fits in with the athlete performance framework	3.86	2.30	
20	everyone to understand what performance health is and isn’t	3.79	2.43	
31 ^$^	vested interests attached to outdated health models to be identified, and recognized as a conflict to the athletes’ interest	3.52	3.07	
**Cluster 11: Use of data and research**		**3.75**	**3.04**	
49	adequate time to implement and evaluate	4.14	3.57	
54	athlete health (e.g., injury, illness, mental health occurrences) to be monitored and reported regularly (e.g., annually) so Executive understands the health status of athletes and where improvements can be made	4.10	2.68	
51	health literacy [how people access, understand and use health information in ways that benefit their health]	4.03	2.71	
42	decisions that are supported by evidence	3.97	2.93	
30	collecting the right data	3.79	2.96	
64	using research to change the way we work	3.69	3.21	
28^^^	coaches to actively implement wellbeing tools provided by the Australian Institute of Sport	3.52	2.93	
36	a data governance framework	3.45	3.25	
61 ^@^	having a research team engage with the athletes as a source for inspiration to what they wish would operate better in their sport	3.03	3.14	
**‘Skilled people’ Domain**		**3.96**	**2.72**	
**Cluster 2: Collaboration and teamwork**		**4.19**	**2.65**	
13	trust in your team	4.66	2.75	
66	the entire team to be in the loop with the same objectives	4.43	2.75	
44	engagement and input across the entire performance support team and coach	4.34	2.79	
3	an interdisciplinary approach [i.e., disciplines working collectively to a common purpose or set goal]	4.31	2.29	
81 ^~^	a fully integrated performance team, including coaches, performance science AND performance health staff	4.31	2.93	
12	a team around the athlete to assist them to become the best athlete who is well-rounded and healthy	4.21	2.43	
7 ^~^	a willingness of all support team members to work with, and enable each other to drive the face-to-face delivery when appropriate	4.17	2.46	
40	an open-minded approach by all members of the performance health team to the evolution of both the areas of health and performance and how their interaction is always changing	4.14	2.71	
76	collaboration across and within the high-performance sport system	4.14	3.00	
95 ^~^	collaborative teams of experts from diverse fields	4.00	2.75	
6	sport science and sport medicine practitioners to support an athlete and coach driven, performance-centered mindset	3.82	2.36	
59	a multidisciplinary approach (i.e., involves team members working independently to create discipline specific care plans)	3.69	2.61	
**Cluster 4: Competency, capacity, and expertise**		**4.14**	**2.76**	
67 ^?^	a strong network of coaches and specialties who the athlete completely trusts	4.41	2.71	
77	the right people, with the right tools, knowledge, and skills to properly address and manage a serious issue	4.37	2.71	
43	world leading coaching and support teams around athletes to provide world leading training environments	4.34	3.07	
9	support teams that have competence or knowledge in injury/illness prevention models, surveillance methods, strategies for action and implementation	4.29	2.79	
58	qualified health staff	4.28	1.89	
17	capacity from service providers to engage in the framework	4.00	2.96	
22	the best practitioners with strong links to coaches	4.00	2.96	
23	the best practitioners with experience and deep understanding of the sport and athletes	4.07	2.86	
96	health literacy across all roles (support team, coaches, and athletes)	4.07	2.89	
87	specialist practitioners to be engaged	3.62	2.75	
**Cluster 12: Roles and responsibilities**		**3.56**	**2.75**	
57	for all staff (including Performance Directors, Coaches, Sport Scientists, Health Professions etc.) to take responsibility for the health and performance outcomes of athletes	4.41	3.11	
46	role clarity among sport science and sport medicine team members	4.07	1.85	
26	physician driven healthcare, supported by allied health practitioners, scientists, and data analytics	3.04	3.11	
80	professionals staying ‘in their lane’ and not trying to be a ‘jack of all trades’	2.69	2.89	
**‘Leadership’ Domain**		**3.98**	**2.79**	
**Cluster 10: Leadership**		**3.98**	**2.79**	
82 ^<^	coach engagement	4.50	2.86	
38	a commitment from the highest levels of the organization to ensure athletes progress through their pathway with minimal risk of physical or mental health issues	4.38	3.11	
93	support from management (e.g., Performance Director or equivalent)	4.28	2.89	
41	that leadership understands where it [performance health] fits within an athlete performance support model to ensure it doesn’t overrun the High-Performance Sport System	3.97	2.68	
55 ^<^	Chief Medical Officers to act as stewards of health	3.48	2.75	
85	to be driven by the coach	3.31	2.46	
**For all statements**		**4.05**	**2.71**	

^+^ Statement 14 was reassigned from Cluster 9 to Cluster 8, * Statements 32 and 69 were reassigned from Cluster 1 to Cluster 3, ^#^ Statements 45 and 89 were reassigned from Cluster 1 to Cluster 6, ^Statement 28 was reassigned from Cluster 6 to Cluster 11, ^!^ Statements 56 and 10 were reassigned from Cluster 10 to Cluster 9, ^$^ Statement 31 was reassigned from Cluster 10 to Cluster 6, ^@^ Statement 61 was reassigned from Cluster 5 to Cluster 11, ^?^ Statement 67 was reassigned from Cluster 12 to Cluster 4, ^~^ Statements 95, 81, and 7 were reassigned from Cluster 12 to Cluster 2, ^<^ Statements 55 and 82 were reassigned from Cluster 4 to Cluster 10, ^>^ Statements 75 and 4 were reassigned from Cluster 4 to Cluster 5.

## Data Availability

The data presented in this study are available on request from the corresponding author. The data are not publicly available due to ethical restrictions around participant privacy.

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
