# Peer review of "What Contributes to Athlete Performance Health? A Concept Mapping Approach"

_ijerph, 2022, doi:10.3390/ijerph20010300_

Round 1

Reviewer 1 Report

Congratulations on preparing, conducting, and writing this manuscript. It clearly advances basic issues for sports performance. Although it was developed under a certain culture (Australia), it is subject to universalization, within the limits of the qualitative methods employed.

The question of the Australian sports culture, while allowing the realization of such a study, could be considered a limiting factor, as it could be, perhaps, a "filter" regarding the 97 statements  and the importance rating?

Just one consideration: In the Abstract, could be the socioecological levels defined?

Author Response

Response to Reviewer 1 comments

Point 1: Congratulations on preparing, conducting, and writing this manuscript. It clearly advances basic issues for sports performance. Although it was developed under a certain culture (Australia), it is subject to universalization, within the limits of the qualitative methods employed.

Response Point 1: Thank you for taking the time to read our manuscript and for providing feedback. We have addressed each comment individually and have made changes in the manuscript accordingly.

Point 2: The question of the Australian sports culture, while allowing the realization of such a study, could be considered a limiting factor, as it could be, perhaps, a "filter" regarding the 97 statements and the importance rating?

Response Point 2:

Thank you for the feedback which was also identified by Reviewer 2. Changes to the manuscript have been made.

Manuscript: Page 15 Lines 368-370

An additional limitation to be aware of when the international community are interpreting these results is that this study is very specific to the Australian sports culture but could also be reproduced in other high-performance sport systems.

Point 3: Just one consideration: In the Abstract, could be the socioecological levels defined?

Response Point 3:

Thank you for the feedback. We have considered the point regarding the potential to highlight the socioecological models within the abstract and agree that while it could be added in, given the constraints of the abstract size, we feel that we cannot adequately describe each level within the abstract. Given that each level is explicitly detailed within the methods, the reader would be able to gather this information as they read the full manuscript

Reviewer 2 Report

The work correctly describes the contributions of a sports ecosystem to the health of high performance athletes.

The similar design of the study stands out for the methodology used to obtain the results that is developed in other works by the scientific team author of the work in relevant academic publications and for precedents of this type of study also within the scientific literature.

As the authors themselves report, the weaknesses arise in the sample size of the athletes who, although they were motivated with an economic reward, do not represent all the social segments that are intended to be addressed in the study design. The study is very specific in the analysis of a geographical area (Australia) but can be reproduced in any other high-performance sports ecosystem.

The design and some results very focused on improving the operation of the system make the academic work correct and very well prepared.

Author Response

Response to Reviewer 2 comments

Point 1: The work correctly describes the contributions of a sports ecosystem to the health of high performance athletes.

The similar design of the study stands out for the methodology used to obtain the results that is developed in other works by the scientific team author of the work in relevant academic publications and for precedents of this type of study also within the scientific literature.

Response Point 1: Thank you for taking the time to read our manuscript and for providing feedback. We have addressed each comment individually and have made changes in the manuscript accordingly.

Point 2: As the authors themselves report, the weaknesses arise in the sample size of the athletes who, although they were motivated with an economic reward, do not represent all the social segments that are intended to be addressed in the study design. The study is very specific in the analysis of a geographical area (Australia) but can be reproduced in any other high-performance sports ecosystem.

Response Point 2: Thank you for the feedback which was also identified by Reviewer 1. Changes to the manuscript have been made.

Manuscript: Page 15 Lines 368-370

An additional limitation to be aware of when the international community are interpreting these results is that this study is very specific to the Australian sports culture but could also be reproduced in other high-performance sport systems.

Point 3: The design and some results very focused on improving the operation of the system make the academic work correct and very well prepared.

Response Point 3: Thank you for the feedback. No change required.
